

# Integrated jerk as an indicator of affinity for artificial agent kinematics: laptop and virtual reality experiments involving index finger motion during two-digit grasping

James Hirose[1], Atsushi Nishikawa[2], Yosuke Horiba[3], Shigeru Inui[3] and Todd C. Pataky[4]

[1] Department of Biomedical Engineering, Shinshu University, Ueda, Nagano, Japan
[2] Department of Mechanical Science and Bioengineering, Osaka University, Suita, Osaka, Japan
[3] Department of Advance Textile and Kansei Engineering, Shinshu University, Ueda, Nagano, Japan
[4] Department of Human Health Sciences, Kyoto University, Sakyo-ku, Kyoto, Japan

Corresponding author
Todd C. Pataky,
pataky.todd.2m@kyoto-u.ac.jp

## ABSTRACT

Uncanny valley research has shown that human likeness is an important consideration when designing artificial agents. It has separately been shown that artificial agents exhibiting human-like kinematics can elicit positive perceptual responses. However the kinematic characteristics underlying that perception have not been elucidated. This paper proposes kinematic jerk amplitude as a candidate metric for kinematic human likeness, and aims to determine whether a perceptual optimum exists over a range of jerk values. We created minimum-jerk two-digit grasp kinematics in a prosthetic hand model, then added different amplitudes of temporally smooth noise to yield a variety of animations involving different total jerk levels, ranging from maximally smooth to highly jerky. Subjects indicated their perceptual affinity for these animations by simultaneously viewing two different animations side-by-side, first using a laptop, then separately within a virtual reality (VR) environment. Results suggest that (a) subjects generally preferred smoother kinematics, (b) subjects exhibited a small preference for rougher-than minimum jerk kinematics in the laptop experiment, and that (c) the preference for rougher-than minimum-jerk kinematics was amplified in the VR experiment. These results suggest that non-maximally smooth kinematics may be perceptually optimal in robots and other artificial agents.

## INTRODUCTION

The uncanny valley is a phenomenon introduced in the robotics literature by Masahiro Mori, explaining that our affinity toward objects generally increases as the object becomes more human-like; however, at a certain human-likeness our affinity drops drastically and we feel a sense of eeriness (*Mori, MacDorman & Kageki, 2012*). With appearance being

quite human but being incompletely so, the viewer senses that something is unnatural and the object is seen as odd, creepy, and/or terrifying.

Kinematics may be important to consider in conjunction with the uncanny valley. Mori pointed out that moving objects can amplify affinity relative to non-moving objects, both in positive and negative senses (such as a corpse being animated becomes much more eerie) (*Mori, MacDorman & Kageki, 2012*). From this idea, one could surmise that different levels of human likeness in the objects' kinematics may modulate uncanny valley responses.

The uncanny valley has arguably been overcome in many applications involving static, lifelike imagery such as computer-generated humans (*Alexander et al., 2010*; *Perry, 2014*), but animated human-like objects (mainly robots) are arguably far less human-like. A variety of research has shown the importance of kinematics in human–robot interactions. For example, it has been shown that both humanoid features and human movements are important factors for facilitating human–robot social interactions (*Kilner, Paulignan & Blakemore, 2003*; *Oztop, Chaminade & Franklin, 2004a*; *Chaminade et al., 2005*; *Glasauer et al., 2010*). Whereas the biological movement in the latter study (*Chaminade et al., 2005*) was from a motion captured trajectory, a separate but similar study (*Huber et al., 2008b*) showed that human–robot interaction improved when the robot moved with minimum-jerk trajectories as approximations to human kinematics (*Flash & Hogan, 1985*), instead of with mechanical trapezoid trajectories. This research suggests that kinematics, and potentially jerk specifically, may interact with the uncanny valley to form an overall affinity for moving artificial agents.

Jerk is the third time derivative of position, and applies to both linear and angular motion. It can be interpreted as the rate of change of acceleration, or perhaps more clearly as proportional to the rate of change of force. Jerk has been shown to be relevant to human–robot interaction (*Huber et al., 2008a*), and theoretical predictions of movement based on a minimum jerk criterion have been shown to qualitatively reproduce many features of human movement (*Flash & Hogan, 1985*; *Furuna & Nagasaki, 1993*; *Viviani & Flash, 1995*). Minimizing jerk is potentially useful movement strategy because high jerk implies rapid force changes and thus both wasted muscular energy and compromised mechanical control. Jerky motion is additionally characteristic of movement disorders (*Latash, 2012*; *Kavanagh, Wedderburn-Bisshop & Keogh, 2016*), implying that lower jerk, if perceivable by humans, may also be perceived as healthier and/or more natural than jerky movements.

The purpose of this study was to quantify the changes in affinity for a moving artificial agent as a function of kinematic jerk. We postulate that jerk may be an important affinity-relevant variable based on previous studies which hypothesize that humans follow minimum jerk trajectories (*Flash & Hogan, 1985*) and that smooth trajectory endpoints are associated with greater affinity (*Huber et al., 2008b*). Through this quantification a secondary goal was to find the optimal smoothness, if any, for artificial agent kinematics.

This study used the hand as an artificial agent model because the hand is more recognizable as human-like than a limb, and because using a hand avoids problems associated with uncanny valley-like responses to heads and faces. We aimed to examine our affinity toward robot hand prostheses by measuring affinity as a function of smoothnesses

from very jerky kinematics, jerkier than a robot, to minimum-jerk kinematics. To secure simplicity for this experiment design, and provide functional findings that can be implemented into robotic designs, we aimed for a simple 1-DoF grasp motion. We chose to examine two-digit index-thumb motion as the simplest possible 1-DOF motion that is still recognizable as human-like.

## EXPERIMENT 1: HUMAN AND ROBOT KINEMATICS

The purpose of this experiment was to roughly quantify the expected range of kinematic jerk values for robot and human finger grasping motions. Specifically the human jerk magnitude that was used for kinematic references and basic kinematic motion robot hand device was used to generate the minimum-jerk value and trajectory.

### Methods

A sample of 21 human subjects was tested. All subjects (including experiment 2 and 3) were requited from Shinshu University and provided written informed consent following the procedures of Shinshu University's ethical review board (approval number: 216).

Two main devices were used for this experiment. The first was a six-camera motion capture system (VENUS 3D Flex13, Nobby Tech. Ltd., Tokyo, Japan) which recorded the positions of reflective markers at a sampling rate of 120 Hz. The second device was an open-source robotic hand (HACKberry, exiii Inc., Tokyo, Japan) (Fig. 1). The individual hand segments were 3D-printed using polylactic acid filament and, unlike the original open-source design, only the wrist segments and more distal segments were used (i.e., the forearm segments were not used). Also unlike the original open-source design, which was battery-powered, the device was modified to accept power using a standard wall adapter. Switches on the back of the hand were used to close the fingers into a grasp-like posture, and then return the fingers to the extended posture depicted in Fig. 1.

Reflective markers were placed on the medial side of the human at the following locations: proximal interphalangeal joint (PIP joint), metacarpophalangeal joint (MP joint), and metacarpal bone (MP bone) (Fig. 2). The motion capture system's six cameras were placed around the table, yielding a total capture volume of approximately 50 cm$^3$.

### *Protocol*

A small eraser ($1.0 \times 1.6 \times 4.5$ cm) was placed on the table with its wide face down, and was used both as a reference point and as a guide for the subjects in order to generate a two finger grasp motion. Each participant held their thumb tip close to the eraser, and keeping an 'L' shape with their two fingers, they extended their index finger while facing their hand's dorsal surface upwards. Marker positions were recorded for a minimum of 60 s at 120 Hz. Subjects were informed to grasp the eraser as they normally do and maintain the grasp until they are instructed to release it and return to their initial position. The interval between movements were 5 s and the grasp-release motions were repeated six times.

A similar motion recording was done on the HACKberry device to measure the grasp range and its time of movement. Like the human hand measurement, markers were placed
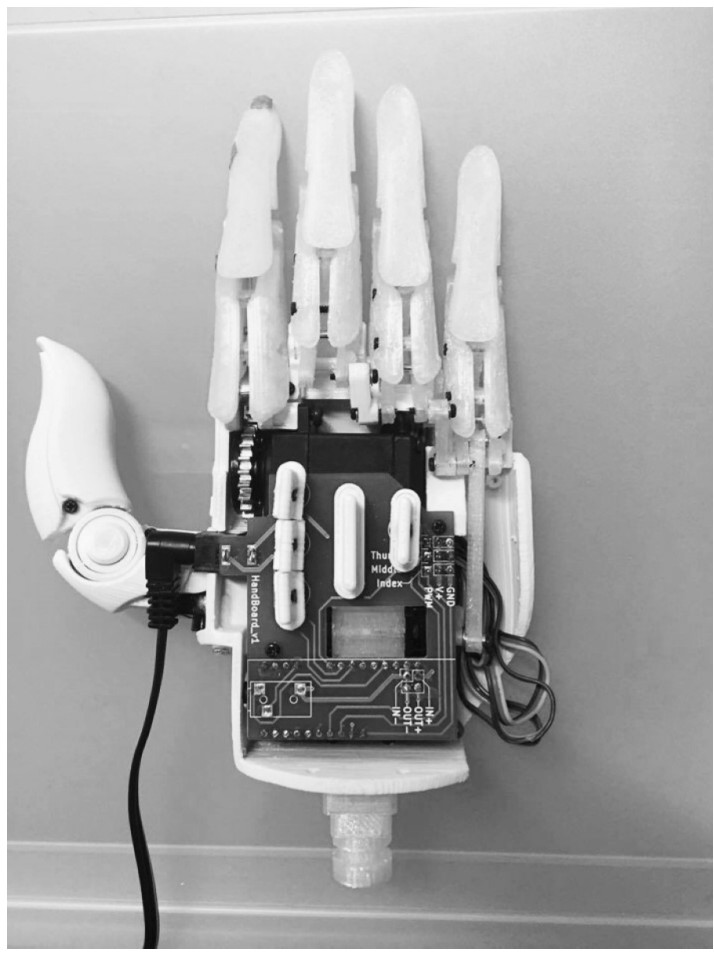

**Figure 1** **Assembled HACKberry system.**

onto the corresponding locations of the HACKberry device and its thumb and index finger will grasp the same object. The grasp motion was measured five times.

## Analysis

Data were processed in Python 3 (Python Software Foundation). From the six grasp-release motions that the human subjects each performed, five grasp trials with the minimal marker swapping were selected for analysis. Each start and end point of the motion were manually determined and the average absolute jerk being calculated as:

$$\text{Average absolute jerk} = \frac{1}{T}\sum_{t=t_0}^{T}\left|\frac{d^3\theta(t)}{dt^3}\right| \tag{1}$$

where $T$ is the total time of the finger motion, $t_0$ is the time at which the finger starts to move, $\theta$ is the angle of the MP joint, and $\theta$'s third derivative was estimated numerically using adjacent time samples. All joint angle data were processed with a low-pass fileter using a fifth-order, zero delay Butterworth filer with a cutoff frequency of 10 Hz.

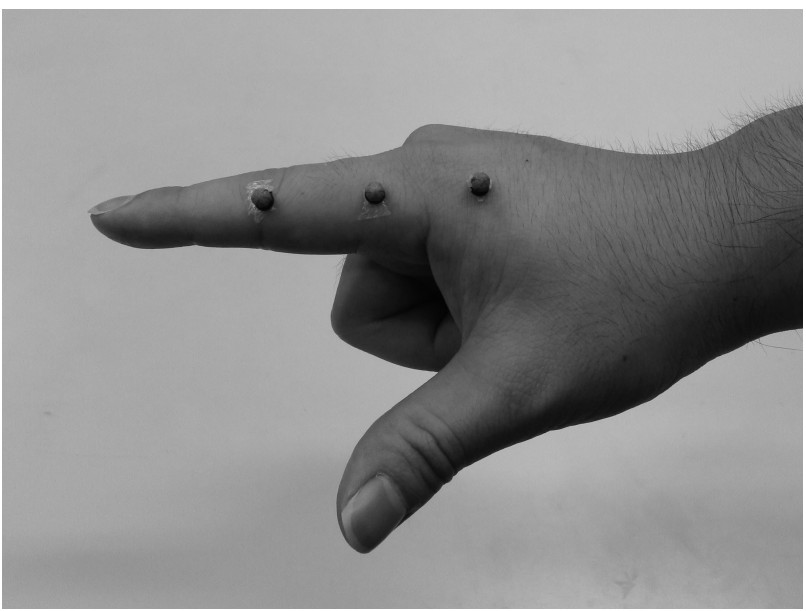

**Figure 2  Marker placements on a human subject hand.**

The HACKberry device was measured for its grasp time length and its angular range by determining the initial and final MP joint angle from the average values. Just like the human subject data, the HACKberry's initial and final point of its motion were manually determined.

Lastly, based on HACKberry's measured data, we created a minimum-jerk trajectory. *Flash & Hogan (1985)* describe a general minimum-jerk trajectory ($x_{\text{minjerk}}$), where an object travels from $x_i$ to $x_f$ in time $= d$ seconds, as:

$$x_{\text{minjerk}}(t) = x_i + (x_f - x_i)\left(10\left(\frac{t}{d}\right)^3 - 15\left(\frac{t}{d}\right)^4 + 6\left(\frac{t}{d}\right)^5\right) \qquad (2)$$

and the measured initial and final points, and the average grasp duration was applied accordingly, and the absolute average jerk magnitude is calculated using Eq. (1).

## Results

As Fig. 3 shows, the average absolute human jerk for the task used in this paper was $0.50 \times 10^5 \text{deg/s}^3$. The HACKberry performed its two-digit 1-degree of freedom (DoF) grasp motion from 161.4 to 96.9 deg in 0.4 s. Thus the average absolute jerk magnitude for the minimum-jerk trajectory for the HACKberry device was $0.20 \times 10^5 \text{deg/s}^3$.

## Brief discussion

Among the results, a subject produced a mean value ($0.190 \times 10^5 \text{deg/s}^3$) that is just under the minimal jerk value ($0.20 \times 10^5 \text{deg/s}^3$). This is because the subject moved its finger slower and/or in smaller pinch angle than the time and/or the travel angle of the minimum-jerk trajectory. Also note that the parameters for the minimum-jerk trajectory was based on the kinematics of the HACKberry robot hand.

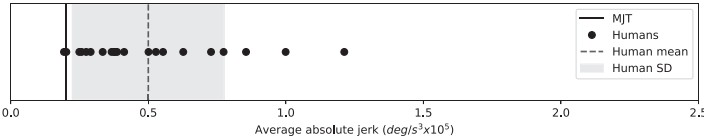

**Figure 3  Absolute human jerk values relative to the minimum-jerk trajectory (MJT).** Each point represents one subject's mean (averaged across five trials). The dashed-line represents the grand mean (across subjects). The shaded area represents standard deviation (*SD*).

Note that this experiment does not seek to identify differences between artificial and human kinematics. An artificial agent was used simply because jerk cannot be controlled precisely in real human motion. The human jerk data from Experiment 1 had just one purpose: to quantify an approximate range of jerk values that can be expected in a 1-DoF two finger grasp motion.

# EXPERIMENT 2: LAPTOP-BASED MOTION PERCEPTION

## Methods

The purpose of this experiment was to quantify subjects' affinity for particular kinematics as a function of kinematic noise amplitude by a one-paired comparison of multiple sample animations on a laptop screen. The animations showed a robot hand (HACKberry device) performing a two-digit 1-DoF grasp motion with different degrees of jerk. We chose to show a robotic-looking hand model instead of a human-looking one because the primary motivation for this study was improving the kinematic quality of artificial agents and not humans. The first part of this Methods section describes how the kinematic samples were created and the second part explains how the software was prepared and used in the experiment.

### *Generating random kinematics with controlled jerk*

Based on the minimum-jerk trajectory generated from Experiment 1, the kinematic trajectory is depicted in Figs. 4C and 4D.

To this minimum-jerk trajectory we added temporally smooth Gaussian noise using the open-source Python package spm1d (*Pataky, 2016*). As compared with temporally rough noise, which is non-physiological, this temporally smooth noise increased jerk in a more natural manner (Fig. 4). We subsequently refer to the amplitude of this smooth, Gaussian noise as 'standard deviation' (*SD*) (Table 1).

Since the constant-weighted Gaussian noise depicted in Figs. 4A and 4B fails to control the initial and final postures, the Gaussian noise was multiplied by a weighting trajectory $w(t)$, which tapered exponentially to zeros at its endpoints, resulting in endpoint-constrained random noise. The final trajectories (Figs. 4A and 4B) were defined as:

$$x(t) = x_{\text{minjerk}}(t) + SD(t)w(t) \tag{3}$$

We created nine random trajectories in total: one minimum-jerk trajectory (*SD*=0) and eight noise-added trajectories with *SD* values: 0.5, 1.0, 1.5, 2.0, 2.5, 3.0, 3.5, 4.0, where

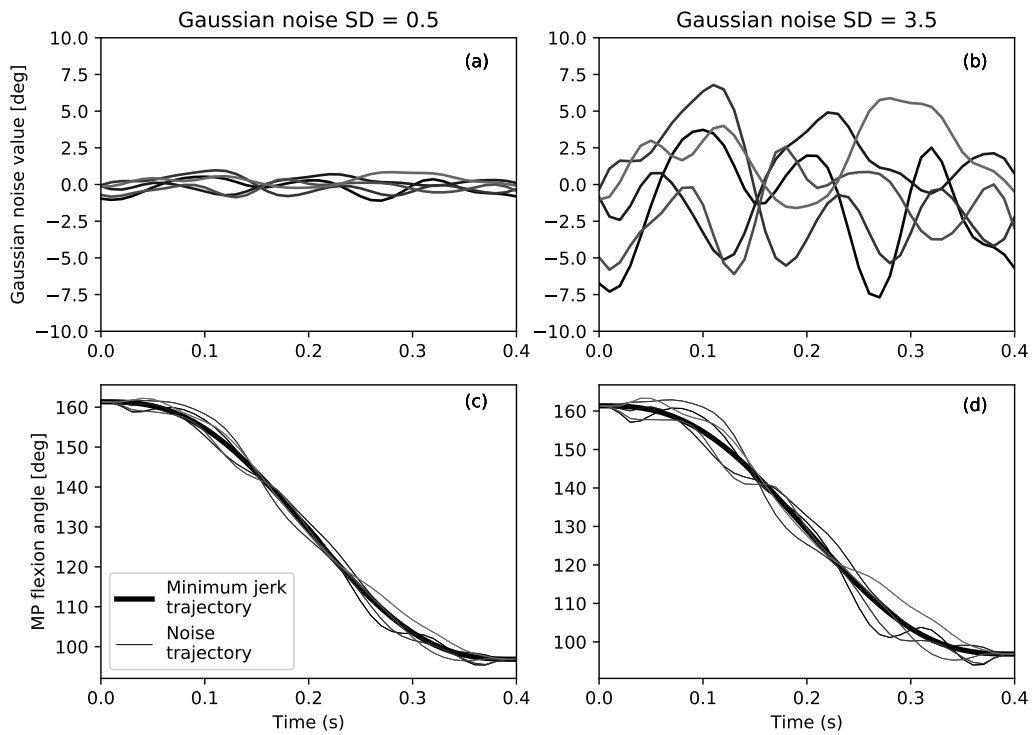

**Figure 4 Example of Gaussian noise added to the minimum jerk trajectory.** Temporally smooth Gaussian noise with two different amplitudes (A) and (B), weighed noise trajectories added to the robot hand's minimum jerk trajectory (fat line). Note that the created trajectories on the bottom graphs are starting and ending at the same points.

**Table 1 Key variable definitions.**

| Variable | Symbol | Description |
|---|---|---|
| Average absolute jerk | (None) | The average value of the third differentiation of the third time derivative of the recorded joint angle trajectory. Due to the nature that jerk value is in constant fluctuation between positive and negative, its value has to be absolute. |
| Gaussian noise | $SD$ | Standard deviation ($SD$) of the Gaussian noise added to the minimum jerk trajectory. $SD = 0$ is the minimum jerk trajectory. Artificially generated trajectories with variance in jerk. |
| Bradley–Terry preference | $p$ | Preference strength, estimated using maximum likelihood, as defined by the Bradley–Terry paired comparison. |

$SD = 0.5$ corresponds to the human $SD$ value measured in Experiment 1. As in Experiment 2 the animations were created using Blender 2.77. Only the index finger was animated, according to the nine trajectories.

Since the aforementioned $SD$ value describes only noise amplitude, which is only indirectly related to kinematic jerk, we also calculated the average absolute jerk following Eq. (1).

### Laptop-based game set up and experiment protocol

The main device used in this experiment was a laptop (MacBook Air, Apple, USA) on which the same animations were displayed but in 2D view. All 22 subjects who participated in this experiment were familiar with laptop use.

Using the Blender Game Engine (Blender Foundation, Amsterdam, www.blender.org), we first developed nine animation videos each based on the generated kinematic samples. Each video showed a HACKberry device an animation of a digital model of the device performing a two-digit grasp: from its starting position (index finger extended) to its movement where its MP joint moved to perform a pinch motion. The animation total length was 2.0 s long (0.8 s of static when extending + 0.4 s motion + 0.8 s of static when grasping). Using those nine animation videos, a game where two of the nine videos with different jerk trajectories were shown side-by-side (Fig. 5), played in loop simultaneously, and players were instructed to "choose the video you find to be most natural" by pressing the left or right key on the laptop's keyboard. After choosing, a new pair of videos was presented and the selection task was repeated until 72 selections were completed (all combinations of the 9 videos, with each left–right pair repeated as right-left). The video pairs were presented in a random order. One game took 15 to 20 min to complete.

## Analysis

As in Experiment 1, data were processed in Python 3.

### Bradley–Terry paired comparison analysis

The goal of this analysis was to quantify preference ($p$). For this purpose we used the Bradley–Terry maximum likelihood estimate of binary preference. The preference for animation $i$ with respect to animation $j$ ($p_{ij}$) was calculated as:

$$p_{ij} = \frac{p_i}{p_i + p_j} \tag{4}$$

where $p_i$ and $p_j$ are the relative selection frequencies of animations $i$ and $j$, respectively, and where $p_i$ and $p_j$ are subject to the constraints:

$$\begin{cases} \dfrac{T_i}{p_i} = n \sum_{j \neq i} \dfrac{1}{p_i + p_j} \\ \sum_{i=1}^{k} p_i = 1 \end{cases} \tag{5}$$

where $T_i$ is the number of times that animation $i$ was chosen over animation $j$.

### Bayesian comparison of preference models

The goal of this analysis was to probabilistically compare different models of the jerk-preference relation. We used the software package PyMC (*Patil, Huard & Fonnesbeck, 2010*) to construct three competing data models (Fig. 6):

Model A: $f(x) = a_0 + a_1 x$

$$\text{Model B:} \quad f(x) = \begin{cases} b_0 + b_1 x, & \text{if } x < b_2 \\ b_0 + b_1 b_2, & \text{otherwise} \end{cases} \tag{6}$$

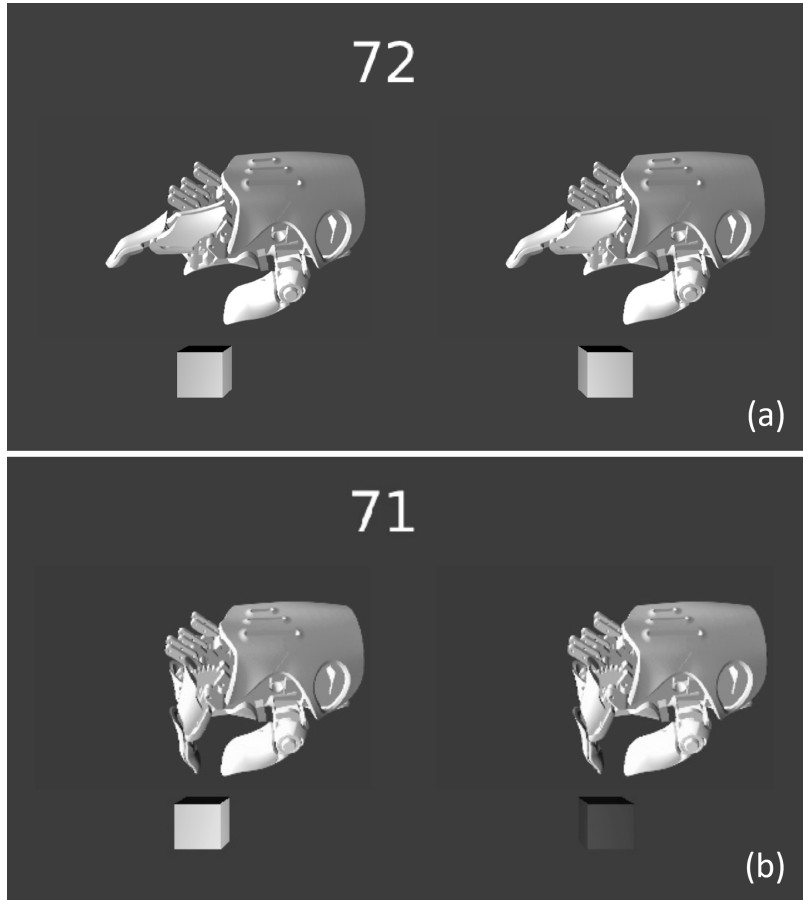

**Figure 5** **Screenshots of the game animation.** Each subject selected one of the hand animations with a left or right arrow key (and one of the two cubes on the bottom of the screen will light up when the key is pressed). The number on the top of the screen indicates the number of remaining pairs to compare. The animations move from an initial open posture (A) to a closed grasp posture (B). Subjects simultaneously observed two animations which had different Gaussian noise '$SD$' (and thus different jerk levels). The animations were randomly selected so that the jerkier animation did not always appear on the same side.

$$\text{Model C}: \ f(x) = \begin{cases} c_0 + c_1 x, & \text{if } x < c_2 \\ c_0 - c_1 x + 2c_1 c_2, & \text{otherwise} \end{cases}$$

where $x$ is absolute average jerk, $a_0$, $b_0$, and $c_0$ are intercepts, and $a_1$, $b_1$, and $c_1$ are slopes. At the transition points $x = b_2$ and $x = c_2$ the modeled behavior changes. The $a$, $b$ and $c$ coefficients are unrelated to variables mentioned in previous sections, and the transition points are restricted into the area of $SD < 1.5$. Models A, B and C represent linear, flattened and reversed preference profiles, respectively (Fig. 6). Finally, the three models will be analyzed in a appropriate range that of $SD < 3.0$.

We separately fit each model to the data using Bayesian inference, with one million iterations, a burn-in of 20,000 iterations, and a thinning rate of six. Relatively weak prior normal distributions were placed on all model parameters. The relative strengths-of-fit were assessed using Bayes factor, with one Bayes factor value computed for each model

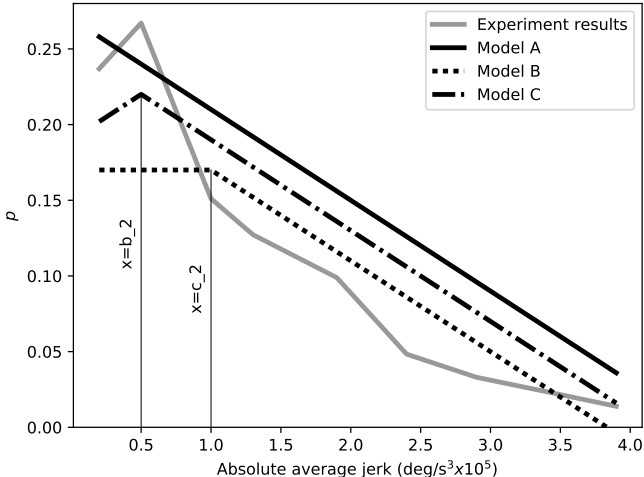

**Figure 6** Examples of the three employed preference models: Model A represents a linear increase in preference with smoothness, Model B represents a flattened preference, and Model C represents a reversed preference.

**Table 2** Bayes factor values comparing preference models within the laptop and VR experiments.

| Models | Laptop | VR |
|---|---|---|
| A vs. B | 8.3 | >1e6 |
| A vs. C | 5.6 | >1e6 |
| B vs. C | 0.7 | 78726 |
| B vs. A | 0.1 | 0.0 |
| C vs. A | 0.2 | 0.0 |
| C vs. B | 1.5 | 0.0 |

pair. Bayes factors were interpreted using previously published guidelines (*Jeffreys, 1998*). Last, sensitivity to the selected prior distributions was assessed by systematically changing the center and breadth of the prior distributions. Results were found to be qualitatively robust to prior distribution adjustments, so sensitivity results are not reported here in interest of space.

## Results

An overall negative correlation between preference and jerk was observed (Fig. 7). An approximately linear trend was observed for absolute average jerk greater than $SD = 0.5$ (jerk $= 0.50 \times 10^5 \text{deg/s}^3$). The minimum jerk trajectory was the second most preferred after $SD = 0.5$, which embodied slighter jerkier kinematics.

Bayesian model comparison found that Models B and C were better fit to the data than Models A (Table 2). According to published interpretation guidelines (*Jeffreys, 1998*), Models B and C were "very strong" fits, and differences between Models B and C were "barely worth mentioning".
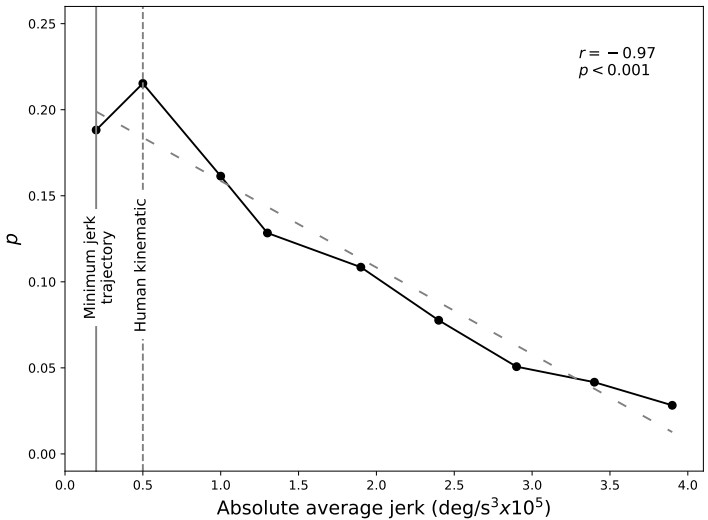

**Figure 7** **Bradley-Terry preference (*p*) for the laptop experiment.** Greater *p* values represent higher preference rankings. The smallest jerk value corresponds to the minimum-jerk trajectory (MJT).

## Brief discussion

The Bradley–Terry paired comparison analysis shows that $SD = 0.5$, the smoothest noise amplitude next to the minimum jerk kinematics and the corresponding to the human kinematic smoothness, was selected as most natural by the subjects. However, whereas the Bayesian analysis shows that both minimum jerk trajectory and $SD = 0.5$ noise amplitude are the suitable candidates for the HACKberry's grasp motion, significance between the two kinematics was not found.

Based on this, we can understand that subjects prefer smoother kinematics, however, it is not clear if subjects find smoother kinematics more natural or human smoothness to be the most natural. Also, even though smoother kinematics were preferred to the jerkier kinematics, the linear increase of the Bradley–Terry preference (*p*) in Fig. 7 suggests that there were subjects that preferred high noise amplitude such as $SD = 3.0$ or $3.5$ above all. From the subject instructions, which pertained only to "naturalness", it is possible that some subjects perceived "natural" motion to encompass jerky motion.

## EXPERIMENT 3: VR-BASED MOTION PERCEPTION

The purpose of this experiment was to examine the same kinematic perception in Experiment 2, but in a virtual reality with increased realism and reduced restraint on movement observation. The idea was that Experiment 2 involved observing grasping motions from a fixed perspective, and that perspective itself could affect motion perceptions. We therefore conducted an additional experiment to test whether freer, interactive observation of the movements in a VR observation would alter the perceptual results we observed in Experiment 2. Subjects wore a VR head mount display with 3D view and 6-DoF head tracking (three translational and three rotational DoF) to be able to observe

the HACKberry models with high realism. Subjects also used two 6-DoF controllers (one for each hand) to control the model postures in the virtual world.

## Methods

Two main devices were used in this experiment: a VR system and a motion controller. The VR system (Oculus Rift, Oculus VR, LLC, Menlo Park, USA) through which custom animations of 3D models were presented in 3D view to subjects as described below. The motion controller (Leap Motion, San Francisco, USA) that subjects used to control objects in the VR environment. Nine subjects participated in this experiment, and all except for one had no or only brief VR experience prior to this experiment. Since the two experiments involved completely different viewing environments, and since it is possible that perceptual metrics had different inter-subject variance in these two experiments, the required numbers of subjects to elucidate a specific effect are not necessarily the same in the two experiments. Additionally, since the perceptual criteria used to judge naturalness may be different between the two experiments, we chose to conduct the results independently, with different subjects and also different numbers of subjects, and to only compare the results qualitatively. The number of subjects for each experiment was decided based on variability estimates from pilot experiments.

Identical to Experiment 2, the VR game presented two side-by-side 3D animations of the HACKberry prosthesis, each repeating a two-digit grasping motion in a loop. Also identical to Experiment 2, the 72 total combinations of the nine grasp trajectories created in Experiment 2 were presented in a random order to each subject. The locations of the animated hands in VR space were translated and rotated accordingly. Thus in the VR world, subjects could rotate the VR hands interactively, freely, and realistically, as if they were the subject's own hands (Fig. 8). They were able to position and rotate the hand models freely. Note that the subjects' finger motion was not tracked, and the HACKberry 3D model's index finger was animated based on the same synthetic kinematics from Experiment 2. Instructions, the video selection task, and game flow proceeded identically to Experiment 2.

## Analysis

As in Experiment 1 and 2, data were processed in Python 3. Also like in Experiment 2, the preference was measured using Bradley–Terry paired comparison analysis, then the Bayesian comparison was done to compare different models of the jerk-preference relation.

## Results

A similar preference shift (a negative trend between preference and jerk) was observed with more of a sigmoidal or s-shaped pattern in the VR experiment. A negative trend in the preference is observed from the $SD = 1.0$ kinematic and reaches little to no preference from $SD = 2.5$ and higher jerk amplitudes. Bayesian model comparison found that Models B and C were better fits to the data than Model A (Table 2). According to published interpretation guidelines (*Jeffreys, 1998*), Models C was a "decisively" better fit than Model B, and differences between Models B and A were "barely worth mentioning".

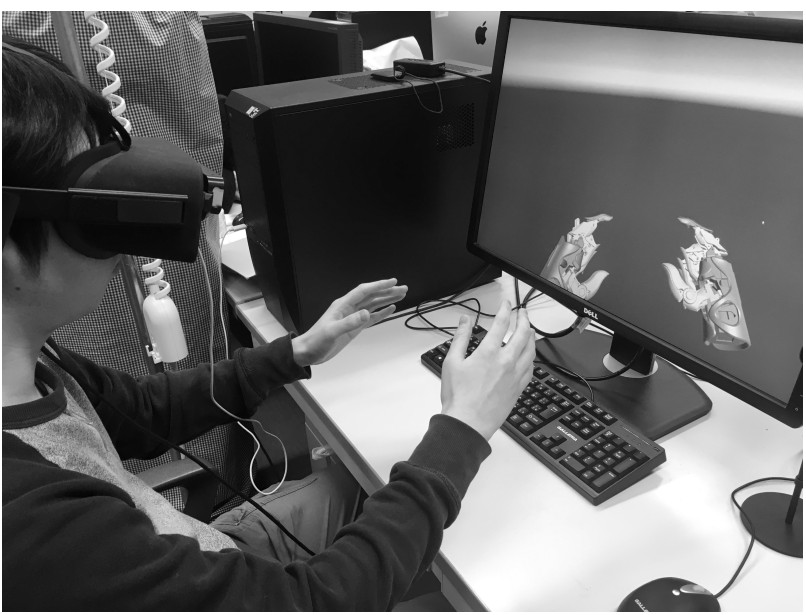

**Figure 8** **VR experiment: Demonstration of a subject observing the robot hand models.** The computer monitor in front of the subject displayed what the subject saw in the VR goggles as reference for the examiner.

## Brief discussion

Both Bradley–Terry paired comparison analysis and Bayesian analysis show that the minimum jerk trajectory is not the optimal smoothness for the HACKberry hand model observed in the VR environrment, and less smooth kinematics are a preferable choice in terms of naturalness. $SD = 1.0$ is ranked the highest for according to the Bradley–Terry analysis, however, the significance between its neighboring kinematic smoothness cannot be found.

From the preference shown in Fig. 9, subjects overwhelmingly find the minimum jerk trajectory and $SD = 0.5$ to 2.0 more natural than the kinematic smoothness of $SD = 2.5$ to 4.0. This can be from the experiment environment where the compared kinematic hand models are moved and observed as if it is the subjects' biological extension, thus the subjects prefer a naturalness relatively more human-like. However, the possibility still remains that an perceptibly optimal smoothness (i.e., noise amplitude) exists that is not the human kinematic smoothness, but specifically for the robotic hand design.

## DISCUSSION

### Main implications

In Experiment 1, we motion captured two-digit 1-DoF grasp motion of human hands and a robot hand device (HACKberry). Human and HACKberry jerk magnitude were calculated and used to formulate an approximate range of jerk values that could be expected in humans and robots. For Experiment 2, we generated computer animations of HACKberry index finger motion in nine steps of jerk over the jerk range calculated in Experiment 1, and
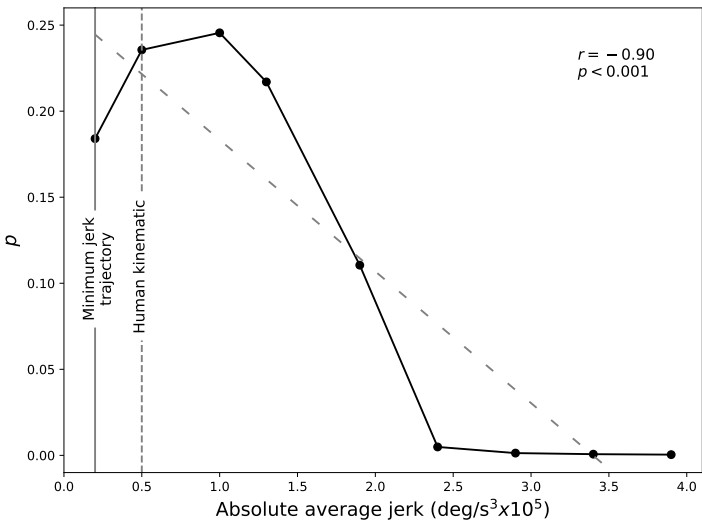

**Figure 9  The Bradley-Terry preference (*p*) for the VR experiment.**  The greater the value, the higher in the preference ranking. All nine subjects' result were used.

we presented these nine animations in pairs to subjects who evaluated their naturalness. Experiment 3 repeated Experiment 2 with new subjects, and in a virtual reality environment where subjects could manipulate the HACKberry animation's 3D positions and rotations using hand-held 6-DoF motion controllers.

This study found that (a) subjects generally preferred smoother kinematics to rough kinematics (Fig. 7), but that (b) subjects did not necessarily prefer the maximally smooth, minimum jerk trajectory to slightly rougher kinematics (Fig. 9). That is, the general trend for preferring smooth vs. rough kinematics did not necessarily apply to the minimum jerk trajectory. Whereas the laptop experiment shows that slightly rougher kinematics ($SD = 0.5$ ($0.50 \times 10^5 \mathrm{deg/s^3}$)) were preferred either equally or more, the VR experiment showed more clearly that subjects preferred slightly rougher kinematics. Both experiments showed qualitatively similar peak preference and also similar preference decline.

Bayesian analyses support these results quantitatively, by suggesting that Models B and C (preference flattening toward maximal smoothness, and preference reduction toward maximal smoothness) both modeled the data better than Model A (linearly increasing preference) (Table 2). This result was amplified in the VR experiment, which also showed that Model C was a considerably better model of the data than Model B. If true, these results suggest that, when designing robots from the perspective of kinematic aesthetics, minimum jerk may not be the optimum solution. This study's results pertain directly only to simple grasping kinematics, but it is possible that these results generalize to broader categories of robotic movements, including general prosthetic hand kinematics, and even general humanoid robot kinematics.

## Qualitative comparisons of the laptop and VR experiments

The laptop and VR experiments yielded similar trends: an overall negative correlation between preference and jerk, but the VR experiment showed that minimum jerk was not preferred to slightly jerkier, human-like kinematics (Figs. 7 and 9).

First, whereas the preference pattern appeared to be approximately linear, or v-shaped for the laptop experiment (Fig. 7), the pattern was sigmoidal, or s-shaped in the VR experiment (Fig. 9). Second, the overall preference weight was different between the two experiments, where slightly smaller preferences were observed in VR experiment. These results suggest that a broader range of low-jerk values may appear natural in VR as opposed to on a 2D computer screen. A study found that subjects wearing a VR head mount display felt more present and immersive than those who were on a 2D monitor (*Gorini et al., 2011*).

Another reason that could have contributed to the weight difference is based on the first-person embodiment perspective nature of the VR experiment; it is possible that the subjects' choice of naturalness became more specific in the VR experiment under a more first-person environment. Both experiments compared the samples under the instruction: "choose the video/hand you find to be most natural", however, to avoid unnecessary biases, no further instructions were given. Each subject therefore judged "naturalness" on unspecified criteria. When witnessing the pairs of grasp animations on the laptop screen (without an attached arm or body), it is possible that some subjects regarded these as non-real, contrived movements, and therefore that it was confusing for them to judge realness. However, in the VR experiment, since subjects could move both themselves and the hands in virtual 3D space, it is possible that they perceived both the hands and the motions as more real, which could have shifted all subjects toward a uniform agreement of their definition of naturalness. Unfortunately, after the experiment subjects were not asked about the criteria they used to judge naturalness. We speculate that the VR environment shifted subjects toward more consistent perceptions do the increased realism of the viewing context.

Finally, it is important to note that the two experiments were conducted independently, with different subjects, and their data were not combined. Nevertheless, it is possible that the two experiments involved similar perceptual mechanisms, so in this Discussion we speculate that (a) some of the perceptual mechanisms may have been common across the two experiments, and that (b) the VR experiment may have enhanced particular perceptual constructs, and that this resulted in reduced inter-subject perceptual variability. We have no direct proof of these speculations; these interpretations are instead hypotheses that are consistent with our data, and that require further testing.

## Relation to uncanny valley literature

Mori hypothesized that the uncanny valley amplifies when objects are in motion (*Mori, MacDorman & Kageki, 2012*). Subsequent studies have shown the negative affinity does not become negatively enhanced as predicted, instead affinity increases in the positive direction (*Piwek, McKay & Pollick, 2014*; *Thompson, Trafton & McKnight, 2011*). Other studies similarly found that subjects interacting with humanoid robots with minimum jerk

trajectories led to cognitive improvements (*Chaminade et al., 2005*; *Huber et al., 2008b*), showing that kinematically natural movement, in this case the minimum jerk trajectory, improves the overall affinity and task performances in human–robot interactions.

This study's findings agree with the fundamental finding that minimum jerk induces greater affinity, but unlike previous studies this study also explored intermediary jerk values. Results suggest that adding slight kinematic noise to robot's minimal jerk movements mimicking, or close to human jerk magnitude may be preferable to the minimum jerk trajectory.

Last, although Mori's uncanny valley findings are generally repeatable, it is notable that uncanny valley effects can depend on the nature of the objects being viewed, and also that uncanny valley existence is not supported in some previous human face viewing experiments (*Seyama & Nagayama, 2007*; *MacDorman et al., 2009*; *Looser & Wheatley, 2010*). Although these studies cast doubt on the universality of the uncanny valley, a separate study found an uncanny valley for hand appearance (*Poliakoff et al., 2013*). In the experiment, subjects rated prosthetic hands with human skin texture as more eerie than mechanically exposed robot hands and actual human hands. This suggests that, although the uncanny valley response can apparently be suppressed in certain circumstances, it can still emerge for non-face body parts. This finding, coupled with the results of the current study, suggest that both object appearance and its kinematics may conspire to alter the uncanny valley response, or more generally, that human likeness and kinematics may interact to form an affinity landscape.

## Limitations

The first limitation of this study was kinematic simplicity. This study involved movement of just a single kinematic degree of freedom (DOF) (the MP joint of the index finger). This was done partly to simplify the experiment and the analysis, but it was also done deliberately as a baseline kinematics test; if non-trivial affinity responses can arise during single DOF kinematics, it would suggest that single DOF is a suitable paradigm for studying at least some aspects of human affinity for robotic kinematics. Furthermore, a similar single-DOF mechanism is used in other prosthesis such as iLimb and Bebionic (see review *Belter et al., 2013*), implying that single-DOF affinity results may have practical implications for currently existing prosthetic control. However, the affinity effects of multi finger kinematics are not clear in this study.

A second limitation was non-varying robot appearance. Reflecting on the studies from Kilner et al. and Kupferberg et al., the humanoid form robot used by Kupferberg et al. caused an improvement in human–robot interaction whereas the industrial robot arm used by Kilner et al. did not. This implies that an element of human-likeness of the robot, such as the appearance, can apparently cancel the effects of kinematic smoothness. We hope to explore this in future studies by systematically varying both human-likeness and kinematic smoothness.

A final limitation may have been movement speed. Separate research on hand prostheses suggests that users have preferred finger flexion speeds, with all female and 76% of male responders describing the grasping speed of their prosthesis as "too slow". A literature

review shows that typical prosthetic index fingers move at a grasping speed of 45.8–103.3 deg/s (medium 91.1 deg/s) (*Belter et al., 2013*). The animations in our study followed out the flexion speeds of 155 deg/s, which are closer to human's everyday pick up task speeds of 172–200 deg/s (*Suteanu et al., 2003*; *Heckathorne, 1992*), and met the suggested criteria of 115 deg/s stated in the aforementioned review.

### Further studies and potential applications

These results could potentially be used as a guideline for tuning the jerk amplitude of a robotic movement. Many robotic trajectory planning applications employ minimum jerk trajectories (*Kyriakopoulos & Saridis, 1988*), but this study's results suggest that minimum jerk may not be optimal if the goal is to maximize the perception of naturalness. Here, as we only used one robot hand in animations, we are unable to make conclusions regarding human-like vs. robot-like appearance. If the goal is to maximize perceptions of naturalness across multiple appearances, future work should consider multiple levels of human-likeliness and the relation amongst appearance, kinematics and affinity. Another factor that requires further study is integrated jerk control in kinematic chains, including multi-joint fingers, and whether kinematics need to be planned at the joint level in order to achieve optimal affinity.

Since it has been reported that human–robot interactions can be improved by implementing the minimum-jerk profile (*Huber et al., 2008a*), it is conceivable that further improvement could be achieved by affinity-maximized kinematics as opposed to jerk-minimized kinematics.

## SUMMARY

This study investigated the relation between kinematic jerkiness and observer preference in single-digit, single degree-of-freedom motion using a robotic hand model. Results suggest an overall negative correlation between absolute average jerk and preference, but also that slightly jerkier kinematics—similar to human jerk levels—may be preferable to minimum jerk. Jerk, and particularly non-minimal jerk, may be an affinity-relevant variable for arbitrary artificial agent kinematics.

### Funding

This work was supported by a Grant-in-Aid for the Shinshu University Advanced Leading Graduate Program by the Ministry of Education, Culture, Sports, Science and Technology (MEXT), Japan, and by Kiban B Grant 17H02151 from the Japan Society for the Promotion of Science. There was no additional external funding received for this study. The funders had no role in study design, data collection and analysis, decision to publish, or preparation of the manuscript.

### Grant Disclosures

The following grant information was disclosed by the authors:

Ministry of Education, Culture, Sports, Science and Technology (MEXT), Japan.
Japan Society for the Promotion of Science: 17H02151.

## Competing Interests

The authors declare there are no competing interests.

## Author Contributions

- James Hirose conceived and designed the experiments, performed the experiments, analyzed the data, prepared figures and/or tables, authored or reviewed drafts of the paper, and approved the final draft.
- Atsushi Nishikawa and Todd C. Pataky analyzed the data, authored or reviewed drafts of the paper, and approved the final draft.
- Yosuke Horiba analyzed the data, authored or reviewed drafts of the paper, advised the statistical analyses, and approved the final draft.
- Shigeru Inui analyzed the data, authored or reviewed drafts of the paper, reviewed the design of the VR pilot experiment, and approved the final draft.

## Human Ethics

The following information was supplied relating to ethical approvals (i.e., approving body and any reference numbers):

This study was approved by the ethical review board of Shinshu University (approval number: 216). All subjects involved in this study provided informed consent following the ethics board-approved procedures.

## Data Availability

Code and data used for this experiment, including statistical analyses, are available at GitHub: https://github.com/JamesShinshu/RS-OpenScience-Codes-and-Data.

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
