# Peer review of "Integrated jerk as an indicator of affinity for artificial agent kinematics: laptop and virtual reality experiments involving index finger motion during two-digit grasping"

_PeerJ, doi:10.7717/peerj.9843_

## Round 0.1 · original submission · Major Revisions

Your manuscript has now been seen by 2 reviewers. You will see from their comments below that while they find your work of interest, major points are raised. We are interested in the possibility of publishing your study, but would like to consider your response to these concerns in the form of a revised manuscript before we make a final decision on publication. We therefore invite you to revise and resubmit your manuscript, taking into account each of the points raised. Please highlight all changes in the manuscript text file.

Reviewer 1 ·

Basic reporting

This manuscript describes preference in the perception of motion rendered on a robotic hand. The scientific approach is valid but the quality falls short of usual standards, and is not sufficient to support the claims of the title and abstract.

Experimental design

- The two experiments, screen and VR, don’t test the same context, and the authors didn’t take into consideration that they shouldn’t be compared directly without discussing these differences. In particular, the n is different but the authors claim it is not an issue (as small n yield false negative and here we have only positive results) yet don’t mention it when making claims about the differences between real screen and VR (p12, l48). This aspect (how is the motion presented?) is not informative for the main argument of the paper regarding motion preference and almost invalidates the other results.

Validity of the findings

- Most important, claiming a preferences in kinematics for artificial agents on the basis of a single experiment with a very simple action (grasp, but mostly transport cf point 1), on one robotic hand with a sample of 22 subjects and 9 kinematics is simply insufficient nowadays, where, in particular for behavioral studies, tens of participants undergo a large number of conditions in order to support the conclusion. In the present case, for example, are particularly missing a larger number of different kinematics in a second experiment, a larger sample to adequately assess inter individual variance, a replication with other robots and actions, a direct comparison with the same actions performed by a human hand with the kinematics of a human hand manipulated along the same dimensions to demonstrate the effect is specific to the robotic hand or simply a feature of the normal action perception system that generalizes to artificial agent. These are just a few examples of the comparisons that would be required to support convincingly the main argument of the manuscript.

Additional comments

This manuscript describes preference in the perception of motion rendered on a robotic hand. The scientific approach is valid but the quality falls short of usual standards, and is not sufficient to support the claims of the title and abstract.

- The article talks about kinematics of grasp, but the three markers are on the index finger and none on the thumb, so it is already a degraded recording of a grasp (transport? index flexion?), and not a two-digit grasping as claimed in the title.

- The two experiments, screen and VR, don’t test the same context, and the authors didn’t take into consideration that they shouldn’t be compared directly without discussing these differences. In particular, the n is different but the authors claim it is not an issue (as small n yield false negative and here we have only positive results) yet don’t mention it when making claims about the differences between real screen and VR (p12, l48). This aspect (how is the motion presented?) is not informative for the main argument of the paper regarding motion preference and almost invalidates the other results.

- The explanations of the differences in results in Table 2 are not convincing. Actually, the RESULTS of Table 2 are not the same as in the previous version of the manuscript I reviewed, which is VERY puzzling:
Current:
Bvs.A 0.7 78726
Bvs.C 0.1 0.0
Previous:
C vs. B 0.7 78726
A vs. B 0.1 0

- Most important, claiming a preferences in kinematics for artificial agents on the basis of a single experiment with a very simple action (grasp, but mostly transport cf point 1), on one robotic hand with a sample of 22 subjects and 9 kinematics is simply insufficient nowadays, where, in particular for behavioral studies, tens of participants undergo a large number of conditions in order to support the conclusion. In the present case, for example, are particularly missing a larger number of different kinematics in a second experiment, a larger sample to adequately assess inter individual variance, a replication with other robots and actions, a direct comparison with the same actions performed by a human hand with the kinematics of a human hand manipulated along the same dimensions to demonstrate the effect is specific to the robotic hand or simply a feature of the normal action perception system that generalizes to artificial agent. These are just a few examples of the comparisons that would be required to support convincingly the main argument of the manuscript.

·

Basic reporting

Introduction
The introduction is generally clear and concise. I think there should be more background information about what jerk is and how it is related to human movement (e.g. what is the normal profile for humans compared to robots).

Towards the end of the experiment, please provide an overview of all the experiments (e.g. that you first collect kinematic information, add this to an animation based on a recorded robot model…..etc). Also, more rationale needs to be added for experiment 3, which seems to cover perspective and embodiment. This will enable the reader to more easily process the methods section.

References to some materials are missing (states (see §)) and some of the figures are not referenced in the text

Experimental design

The study is within the scope of the journal.
The research question is well defined and relevant - this is an unexplored area and i like the quantitative and detailed approach the authors use.

Methods
The layout of the methods and analysis is unclear. My questions below could probably be answered if the methods were structured so that each experiment is presented separately, followed by a general discussion. For example:
Experiment 1: methods, analysis, results, brief discussion
Experiment 2: methods (making use of exp 1 data), analysis, results, brief discussion. Make it clearer that kinematics were added to an animation of the hackberry robot
Experiment 3: methods, analysis, results, brief discussion

Equipment – this states that there were four main devices, but there are actually five

Experiment 1 – how is the data used? The rationale for this needs to be clearer and the use of the kinematics could be added to experiment 2 methods (developing the stimuli). How were those specific kinematics chosen?
Experiment 2 – indicate that the stimulus was robotic looking (based on the hackberry model)
Experiment 3 – it is not clear when or why the participants would move the hand stimuli. How are they stopped from moving them during each trial? Overall, there needs to be more description of the procedure and rationale for this experiment.

Validity of the findings

Results
Experiment 2 and 3 – please add statistics for correlations (r/p values)

Discussion
Comparing laptop and VR experiments. The following lines need to be backed up in the results section. The linear trend/correlation is mentioned for the laptop experiment but not for the VR experiment . Please document these differences statistically in the results (and /or descriptively).

“but the VR experiment showed that minimum jerk was not preferred to slightly jerkier, human like kinematics (figures 8 and 9). First, whereas the preference pattern appeared to be approximately linear, or v-shaped for the laptop experiment (figure 8), the pattern was sigmoidal, or s-shaped in the VR experiment (figure 9). “

This sentence does not make sense:
“In the experiment, subjects rated prosthetic hands with human skin texture as more eery than all of: mechanically exposed robot hands, prosthetic hands with human skin texture, and actual human hands.”

Please provide a little more discussion around the potential use of the findings (for robotics/prosthetic or neuroscience research). The authors briefly mention previous findings that have used different appearances, but a more detailed discussion on how appearance and kinematics might interact would be valuable.

---

## Round 0.2 · accepted · Accept

Thank you for the detailed response letter. We are delighted to accept your manuscript for publication.

·

Basic reporting

The authors have answered my comments

Experimental design

The authors have answered my comments

Validity of the findings

The authors have answered my comments